# Ebola virus disease in the Democratic Republic of the Congo, 1976-2014

Alicia Rosello[1,2]*[†], Mathias Mossoko[3†], Stefan Flasche[4], Albert Jan Van Hoek[4], Placide Mbala[5], Anton Camacho[4], Sebastian Funk[4], Adam Kucharski[4], Benoit Kebela Ilunga[3], W John Edmunds[4], Peter Piot[4], Marc Baguelin[1,4]*, Jean-Jacques Muyembe Tamfum[5]

[1]Public Health England, London, United Kingdom; [2]University College London, London, United Kingdom; [3]Direction de lutte contre la maladie, République Démocratique du Congo Ministére de la santé Publique, Kinshasa, Democratic Republic of the Congo; [4]London School of Hygiene and Tropical Medicine, London, United Kingdom; [5]Institut National de Recherche Biomédicale, Kinshasa, Democratic Republic of the Congo

**Abstract** The Democratic Republic of the Congo has experienced the most outbreaks of Ebola virus disease since the virus' discovery in 1976. This article provides for the first time a description and a line list for all outbreaks in this country, comprising 996 cases. Compared to patients over 15 years old, the odds of dying were significantly lower in patients aged 5 to 15 and higher in children under five (with 100% mortality in those under 2 years old). The odds of dying increased by 11% per day that a patient was not hospitalised. Outbreaks with an initially high reproduction number, R (>3), were rapidly brought under control, whilst outbreaks with a lower initial R caused longer and generally larger outbreaks. These findings can inform the choice of target age groups for interventions and highlight the importance of both reducing the delay between symptom onset and hospitalisation and rapid national and international response.

*For correspondence: alicia.rosello.13@ucl.ac.uk (AR); marc.baguelin@lshtm.ac.uk (MB)

[†]These authors contributed equally to this work

**Competing interests:** The authors declare that no competing interests exist.

## Introduction

Ebola virus disease (EVD) outbreaks are rare and knowledge of the transmission and clinical features of this disease is sparse. As of May 2015, the devastating outbreak in West Africa has resulted in more than ten times the number of cases reported in all previous outbreaks and will ultimately provide improved insights into EVD. Here, for the first time, all the databases from EVD outbreaks in the Democratic Republic of the Congo (DRC) have been cleaned and compiled into one anonymised individual-level dataset (See *Supplementary file 1*). The data provided are an invaluable addition to the West Africa data and will allow a more complete picture of the disease. The DRC is the country that has experienced the most outbreaks of EVD. Since the virus' discovery in 1976, there have been six major outbreaks (Yambuku 1976, Kikwit 1995, Mweka 2007, Mweka 2008/2009, Isiro 2012, and Boende 2014) and one minor outbreak (Tandala 1977) reported in the DRC, four in the northern Equateur and Orientale provinces and three in the southern provinces of Bandundu and Kasai-Occidental (*Figure 1*). Some of these have been described in the literature (*World Health Organization, 1978*; *Heymann et al., 1980*; *Khan et al., 1999*; *Muyembe-Tamfum et al., 1999*, *2012*; *Maganga et al., 2014*). However, the individual-level data and corresponding lessons from these outbreaks have not been collated or made publicly available; by doing so, we aim to permit a more powerful statistical analysis and a fuller understanding of the disease. The end of the most recent outbreak in the DRC was declared on the 21st of November 2014. This provides an

**eLife digest** Ebola virus disease commonly causes symptoms such as high fever, vomiting, and diarrhoea. It may also cause muscle pain, headaches, and bleeding, and often leads to death.

There have been seven outbreaks of Ebola virus disease in the Democratic Republic of the Congo (DRC) since 1976. The DRC is the country that has had the most outbreaks of this disease in the world. The most recent outbreak in the DRC was in 2014; this was separate from the outbreak that started in West Africa in the same year. Rosello, Mossoko et al. have now compiled the data from all seven of the outbreaks in the DRC into a single dataset, which covers almost 1000 patients.

Analysing this data revealed that people between 25 and 64 years of age were most likely to be infected by the Ebola virus, possibly because most healthcare workers fall into this category. Age also affected how likely a patient was to die, with those aged under 5 and over 15 more likely to die than those aged between 5 and 15. Delaying going to hospital once symptoms had started, even by one day, also increased the likelihood of death.

Rosello, Mossoko et al. also examined the Ebola virus effective reproduction number, which indicates how many people, on average, an infected person passes the virus on to. Outbreaks that initially featured viruses with a reproduction number larger than three tended to be stemmed quickly. However, when the reproduction number was lower, national and international organisations were slower to respond to the signs of the outbreak, leading to outbreaks that lasted longer.

Further research is needed to understand why the likelihood of death is different for different age groups and to investigate the effect of the different routes of transmission of the virus on interventions such as vaccination.

unparalleled opportunity to assemble all the information gathered about EVD in the DRC through almost four decades, learn from the Congolese experience with this disease, and compare the features of EVD in DRC with the epidemic that has had such a devastating effect in West Africa.

## Results

During the last 38 years, 1052 cases of EVD have been reported in the DRC, of which 996 are reported in this dataset. The geographical context, historical timeline, and main characteristics of these outbreaks are depicted in *Figure 1* and *Table 1*. A detailed account of the outbreaks can be found in Appendix 1, Section B. The early accounts of all outbreaks except for Mweka 2007 involved a healthcare facility. The direct epidemiological link between index cases (when known) and animal reservoirs has not been found for any of the outbreaks. The lack of systematic surveillance together with the presence of diseases with similar symptoms allows EVD cases to go unnoticed for long periods of time. A repository of the interventions that led to the control of the outbreaks is outlined in *Supplementary file 2*. *Table 2* summarises the number of cases and deaths reported in each outbreak.

### Case demographics

The number of cases and case-fatality ratios (CFRs) varied greatly between outbreaks (*Table 2*). It can also be observed that laboratory confirmation became more readily available over time. Across all outbreaks, 57% of cases were female (95% CI = 53.9–60.1). In the second Mweka outbreak and in the Isiro outbreak, more than 70% of cases were females. However, in the other outbreaks, the percentage of females was lower (53–59%). When comparing the probable and confirmed cases by age with the overall DRC population (*Figure 2*), we observed a high concentration of cases in the 25–64 age category compared to the baseline population. This might be because at this age individuals are more likely to be carers. The occupation was only recorded during three outbreaks: Kikwit, Boende and Isiro. During Kikwit, 23% (73/317) of cases were known healthcare workers (HCWs) and 0.6% (2/317) were possible HCWs. During Boende, the occupation was recorded for 85% (58/68) of cases. 14% (8) were known HCWs and 3% (2) were possible HCWs. During Isiro, occupation was reported for 94% (49/52) of cases. 27% (13) were HCW. Although occupation was not recorded on an individual level, during Yambuku, 13 of the 17 Yambuku Hospital workers contracted EVD (*World Health Organization, 1978*).

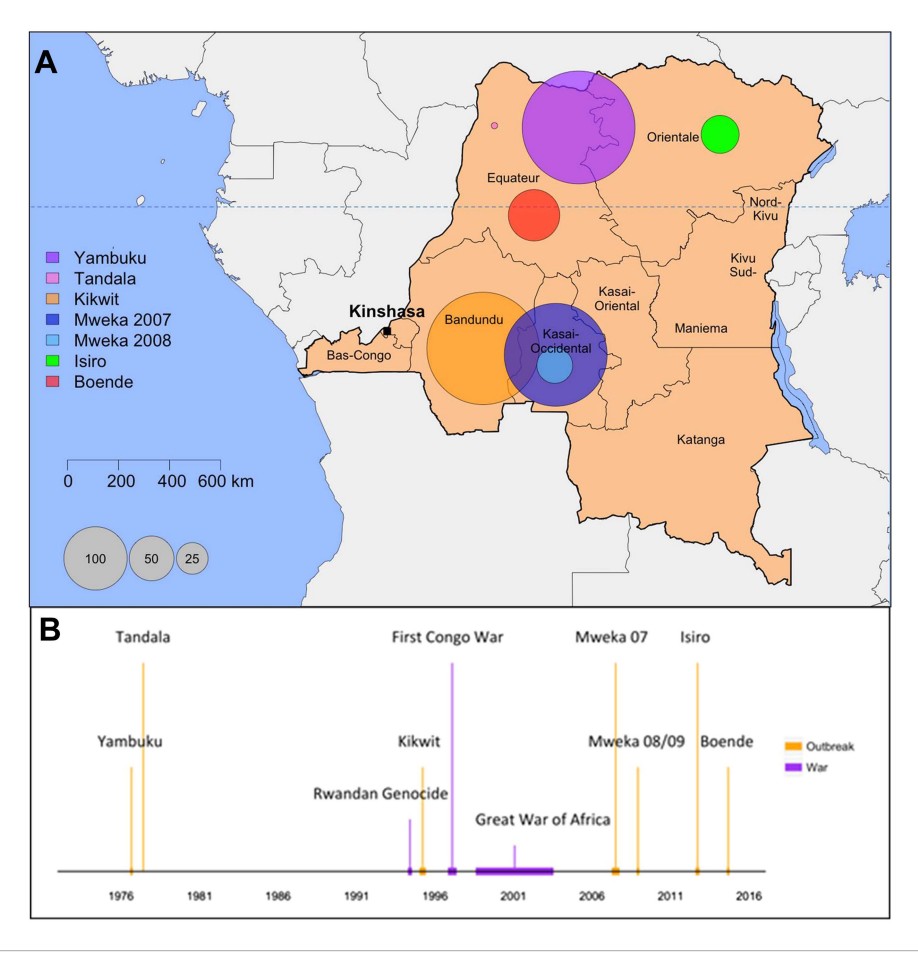

**Figure 1**. Map and historical timeline of the EVD outbreaks in the DRC. (**A**) Map of the Democratic Republic of the Congo (DRC) where the area of the circles are proportional to the number of cases (probable and confirmed) per outbreak. (**B**) The outbreaks (in orange) and relevant wars (in purple) are positioned in time.

## Epidemic curves

The epidemic curves were plotted for the six major outbreaks (*Figure 3*). The date of infection was based on symptom onset when available (701/995). When it was not, hospitalisation dates were used (5/995). In cases where these were also absent (281/995), the notification dates were used as proxy. For Mweka 2007, the date of infection was mostly based on the notification date (98%), whereas in the other outbreaks, infection dates refer to onset of symptoms almost exclusively (>90%). In time, case definitions became more specific. With the exception of Kikwit, in which notification and the closure of healthcare facilities coincided closely in time, outbreaks seemed to peak before major interventions were initiated.

## Symptoms

The proportion of probable and confirmed cases reporting EVD symptoms is shown in *Figure 4*. Overall, the most commonly reported symptom was fever, which was reported by 95% of cases (95% CI = 92.6–97.3%) and at least 90% of cases in every outbreak. Reports of vomiting were also similarly common across all major outbreaks, reported by 75% of cases (95% CI = 69.3–79.2) and between 57% and 76% of cases for all major outbreaks. There was considerable variation in how frequently the remaining symptoms were reported for different outbreaks. In particular, hemorrhagic symptoms were present in 61% (95% CI = 51–71) of cases during Kikwit but only 10% (95% CI = 5–18) during Mweka 2007. The *Bundibugyo ebolavirus* (Isiro outbreak) did not present a symptom profile that was particularly different from that seen for the *Zaire ebolavirus* (all other outbreaks). However, this was difficult to conclude given the large variation between outbreaks.

**Table 1**. Main characteristics of the outbreaks

| | Yambuku | Tandala | Kikwit | Mweka 07 | Mweka 08/09 | Isiro | Boende |
|---|---|---|---|---|---|---|---|
| Ecosystem | Tropical rain forest | Rainforest/ savannah | Urban/peri-urban | Forested savannah | Forested savannah | Tropical area of savannah scattered with gallery forests | Tropical rainforest |
| Inhabitants | Small villages <500 residents | Small village | Villages and city of 200,000 | 170,000 | 170,000 | 700,000 exposed | 250,000 in Boende but most cases living in small villages |
| Start | Aug-76 | Jun-76 | Jan-95 | Apr-07 | Nov-08 | Jun-12 | Aug-12 |
| End | Oct-76 | Jun-76 | Jun-95 | Oct-07 | Jan-09 | Nov-12 | Oct-12 |
| Healthcare facility involved in history | Yambuku Catholic Mission Hospital | – | Kikwit II Maternity Unit and Kikwit General Hospital | – | Kaluamba health centre, injections clandestine nurse from Kaluamba | Chemin de Fer des Uélé clinic (Isiro), Isiro General Reference Hospital | Antenatal care in her village, Miracle centre in Isaka, Lokolia health centre |
| Index case detected? | No | No | Possible: charcoal maker who worked in the forest | No | No | No | No |

## CFRs

The mean CFR overall was 79% (95% CI = 76.4–81.6), but there were significant differences between epidemics and within epidemics over time (*Figure 5* and *Figure 5—figure supplement 1*). The highest average CFR was seen during the first outbreak in Yambuku (mean = 96%, 95% CI = 92.6–97.9 in our subset of 262/318 cases). Kikwit, Mweka 2007, and Boende had high average CFRs ranging from 74% to 78%. During the Isiro and Mweka 2008 outbreaks, the CFR was lower, at 54 and 44% (95% CI = 39.5–67.8 and 26.4–62.3), respectively.

All EVD patients under 2 years of age died (N = 29, *Figure 5—figure supplement 1*). CFRs generally decreased during childhood and then increased again to plateau at around 70–80% in adulthood (*Figure 5—figure supplement 2*). This pattern was less readily observed for the CFRs in the Yambuku outbreak, which remained high and similar for all ages.

**Table 2**. Distribution of cases and deaths by type and overall case-fatality ratios per outbreak

| | Yambuku (1976) | Kikwit (1995) | Mweka (2007) | Mweka (2008/9) | Isiro (2012) | Boende (2014) | All outbreaks |
|---|---|---|---|---|---|---|---|
| Cases (n) | | | | | | | |
| Suspected | – | – | – | – | 0 | 2 | 2 |
| Probable | – | 317 | – | 22 | 16 | 28 | 383 |
| Confirmed | – | – | 24 | 10 | 36 | 38 | 108 |
| Total | 318* | 317 | 264 | 32 | 52 | 68 | 733 |
| Deaths (n) | | | | | | | |
| Suspected | – | – | – | – | 0 | 0 | 0 |
| Probable | – | – | – | 12 | 15 | 28 | 55 |
| Confirmed | – | – | 17 | 2 | 13 | 21 | 53 |
| Total | 280* | 248 | 187 | 14 | 28 | 49 | 526 |
| Case-fatality ratio | | | | | | | |
| (%, 95% CI) | 88* | 78 (73–83) | 74 (68–79) | 44 (26–62) | 54 (39–68) | 74 (62–84) | 79 (76–82) |
| Sex | | | | | | | |
| (% Female, 95% CI) | 59 (53–65) | 54 (48–59) | 55 (48–61) | 72 (53–86) | 77 (63–87) | 53 (40–65) | 57 (54–60) |

Where the distinction between probable, confirmed, and suspected cases was available, the case-fatality ratio and % female were calculated with only probable and confirmed cases. Only cases for which outcomes were reported were included in the case-fatality ratio denominator.

*The values presented in this table for Yambuku were taken from the literature, as our data are a subset of the total cases during the outbreak (262/318).

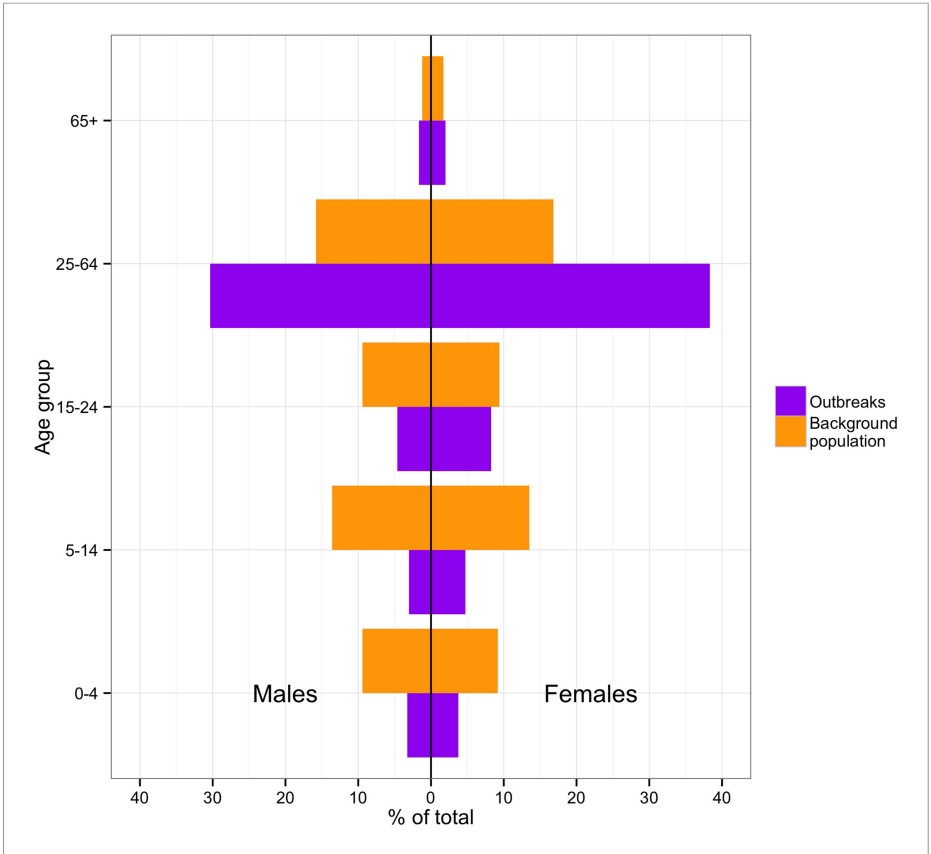

**Figure 2**. Incidence of cases by age and sex in the DRC outbreaks in comparison to the demographics of the national 1975–2010 population.

In the regression model that included the delay between symptom onset and hospitalisation as a factor but excluded three outbreaks for missing data (*Table 3*), the baseline CFR in individuals over 15 years of age during the first month of an EVD outbreak who were admitted to hospital after 0.3 days (the average time from symptom onset to admission to hospital) during the Boende outbreak was 74% (95% CI = 17.8–99.3). The CFR was similar during the Isiro outbreak but was significantly higher during the Kikwit outbreak (94%). The CFR in 0–5 year olds was 76%, and in 5–15 year olds, it was significantly lower at 36%. The odds of dying declined on average by 31% (95% CI = 3.1–52.0) each month after the start of an outbreak and increased by 11% (95% CI = 1.8–20.7%) per day that a symptomatic person is not hospitalised (*Table 3*).

In the regression model that included all major outbreaks, the CFR for individuals over 15 years of age during the first month of the outbreak during the Boende outbreak was estimated at 79% (95% CI = 25.8–99.5). The Yambuku, Kikwit, and Mweka 2007 outbreaks had significantly higher CFRs (96%, 94% and 93%) and the Mweka 2008 outbreak had a significantly lower CFR (48%). 0–5 year olds had significantly higher CFRs (90%) than those over 15 years of age. For the 5–15 year olds, the CFR was significantly lower (57%). The odds of dying declined on average by 35% (95% CI = 22.6–45.9) each month after the start of each outbreak (*Table 4*).

## Reproduction numbers through time

Changes in the effective reproduction number, R, over the course of the outbreaks were plotted in *Figure 6*. In Yambuku, Mweka 2008, and Boende 2014, R dropped below one within 3–5 weeks after the initial case and the outbreak was rapidly brought under control. In these settings, the spread of EVD during the first 2 weeks had been high (R > 3). By contrast, in Kikwit 1995, Mweka 2007, and Isiro 2012, where the initial transmission rate was lower, spread of EVD was sustained for more than 13

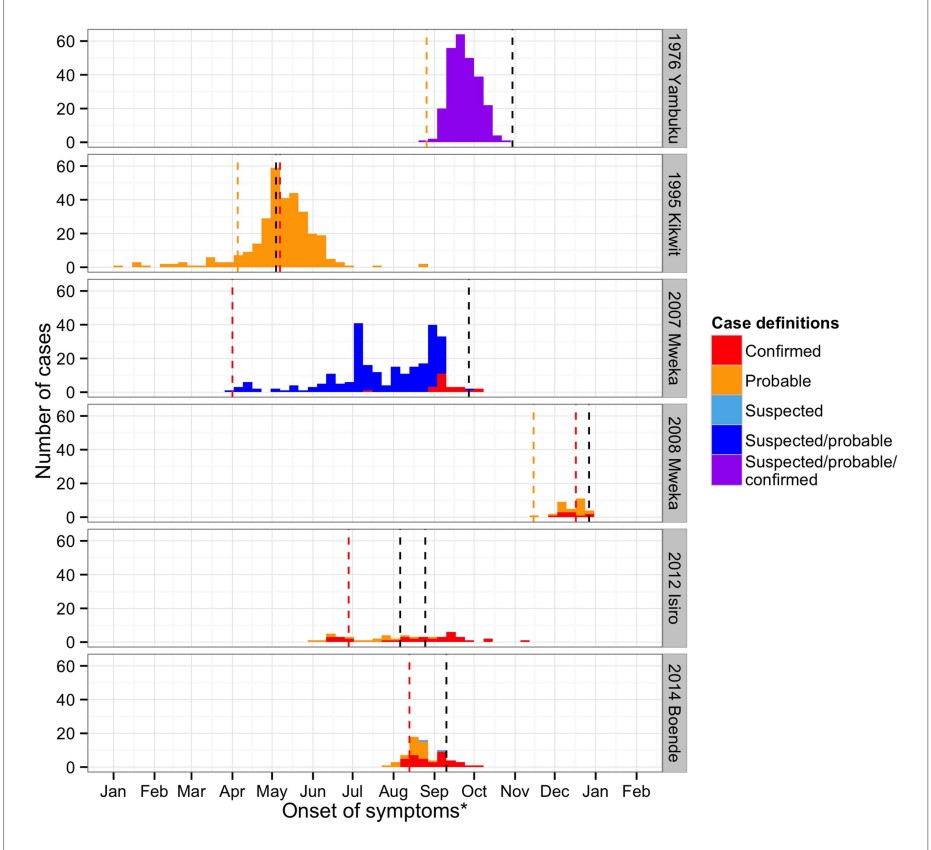

**Figure 3**. Time course of the EVD outbreaks in DRC. Confirmed cases are plotted in red, probable cases in orange, suspected in light blue, cases that were either suspected or probable cases in dark blue, and cases for whom the definition was unknown in purple. The dashed lines represent important events that occurred during the outbreaks (in orange, the first records of the disease, in red, the first notifications, and in black, important interventions carried out). For Yambuku, this was the closure of Yambuku Mission Hospital; for Kikwit, the closure of all hospitals, health centres, and laboratories in the area; for Mweka 2007, the opening of two mobile laboratories; for Mweka 2008, the opening of the first isolation centre; for Isiro, first the opening of the isolation centre and later the opening of the laboratory; and for Boende, the opening of the first isolation centre. Notification dates were when the cases were first notified to the Direction de Lutte contre la Maladie (DLM).

weeks. Overall, we can see that R declines before the major interventions occurred, which could point to behavioural changes that occurred spontaneously in the populations.

## Delays in case detection

The delay distributions from onset of symptoms to notification, from onset of symptoms to hospitalisation, from onset of symptoms to death, length of hospital stay, and from hospitalisation to death were plotted for each outbreak (when available) in *Figure 7*. The largest delays between symptom onset to notification and to hospitalisation were seen during the Kikwit outbreak (12.9 days and 5.0 days, respectively). The largest delay between symptom onset and death and the longest duration of hospitalisation were seen during the Isiro outbreak (11.4 and 8.0 days, respectively). However, this was only recorded for the Kikwit, Mweka 2008, and Isiro outbreaks. The longest delay between hospitalisation and death was observed during the Mweka 2008 outbreak (11.0 days) (*Table 5*).

## Discussion

This article provides for the first time a description and a line list for all outbreaks that have occurred in the DRC. This represents almost 40 years of surveillance data, seven outbreaks, and 996 suspected,

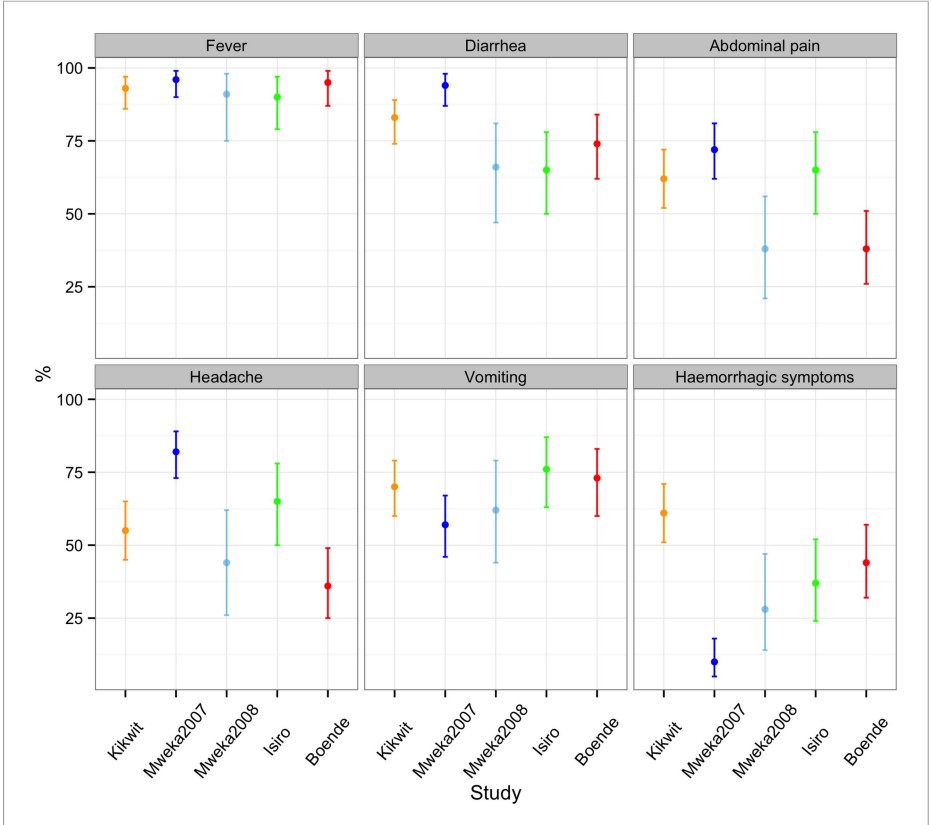

**Figure 4**. Percentage of probable and confirmed cases with abdominal pain, diarrhoea, fever, haemorrhagic symptoms, headache, and vomiting. These were calculated by dividing the number of probable and confirmed cases with symptoms by the number of probable and confirmed cases with symptoms, no symptoms, and blanks for cases for who the presence or absence of at least one symptom was reported. Note that the majority of cases in the Mweka 2007 outbreak were diagnosed a posteriori using recorded symptoms.

probable, or confirmed cases. It is an invaluable resource for studying the epidemiology and clinical features of EVD. We highlight the importance of reducing the delay between symptom onset and hospitalisation, as the odds of dying increase by 11% per day that a patient is not hospitalised. We also observe higher incidence in those between 25 and 64 years of age and a higher CFR in patients under 5 or over 15 years of age than in those between 5 and 15 years old. These trends mirror those observed during the West African outbreak, where cumulative incidence was highest in those between 16 and 44 years of age and CFR progressively dropped from 89.5% in those under 1 year of age to 52.1% in those between 10 and 15 years, to rise again to 78.7% in those over 45 years old (*WHO Ebola Response Team et al., 2015a*). These distinctions could inform the choice of target age groups for interventions such as vaccination.

Another important finding is that during outbreaks with an initially lower reproduction number, R, (≤3) national and international response was slower, outbreaks took longer to control, and (with the exception of Yambuku, where the virus was first discovered) were larger outbreaks than those with initially high R. This occurred during the current outbreak in West Africa, where the basic reproduction numbers for Guinea, Sierra Leone, and Liberia have been estimated at 1.51, 2.53, and 1.59, respectively, and indicates the need for any future EVD to be met with rapid national and international response (*Althaus, 2014*).

Our estimates largely coincide with those recently reviewed in the literature (*Van Kerkhove et al., 2015*). The basic reproduction numbers reported for the Kikwit outbreak (3.00) is comprised in the range found by other studies (1.36–3.65) (*Chowell et al., 2004*; *Ferrari et al., 2005*; *Lekone and Finkenstädt, 2006*; *Legrand et al., 2007*; *Forsberg White and Pagano, 2008*; *Ndanguza et al., 2011*),

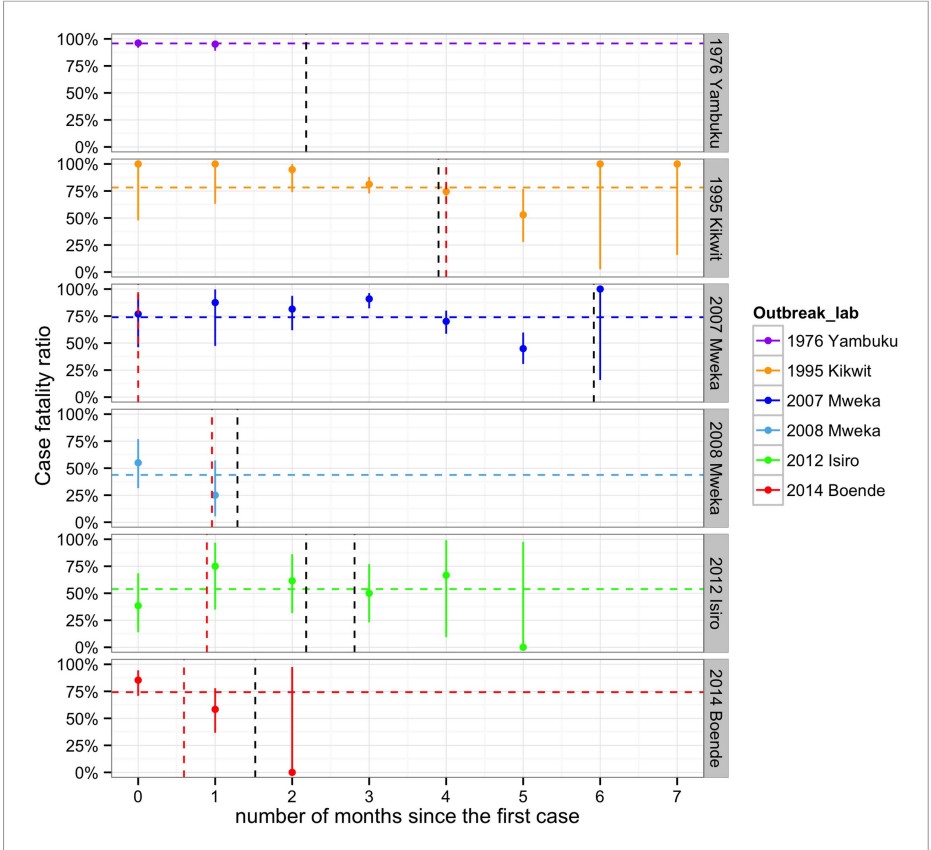

**Figure 5**. Evolving case-fatality ratios with time after the start of the outbreak. Monthly point estimates are presented with 95% binomial confidence intervals. The dashed horizontal line indicates the average case-fatality ratio (CFR) during each outbreak. The vertical dashed lines represent important events that occurred during the outbreaks (in red, the first notifications, and in black, important interventions carried out). For Yambuku, this was the closure of Yambuku Mission Hospital; for Kikwit, the closure of all hospitals, health centres, and laboratories; for Mweka 2007, the opening of two mobile laboratories; for Mweka 2008, the opening of the first isolation centre; for Isiro, first the opening of the isolation centre and later the opening of the laboratory; and for Boende, the opening of the first isolation centre. Notification dates were when the cases were first notified to the DLM.

The following figure supplements are available for figure 5:

**Figure supplement 1**. CFR by age groups for each outbreak.

**Figure supplement 2**. Aggregated CFRs for all outbreaks by age group.

and our estimate for the Yambuku outbreak (5.00) is similar to that reported by Camacho et al. (4.71, range = 3.92–5.66) (*Camacho et al., 2014*). The mean delay of onset of symptoms to hospitalisation and to death estimated here for Kikwit (5.0 and 9.5, respectively) was also similar to that found by other authors (4–5 [*Khan et al., 1999*; *Rowe et al., 1999*] and 9.6–10.1 [*Bwaka et al., 1999*; *Khan et al., 1999*]). Our estimated mean delay of onset of symptoms to death during the Boende outbreak (9.4) was slightly lower than that found by other authors (11.3) but included in their reported range (1–30) (*Maganga et al., 2014*). The delay between hospitalisation and death during the Kikwit outbreak found in the literature (4.6) coincided with our estimate (4.5) (*Khan et al., 1999*). In addition, our estimates of the overall CFR for Kikwit and Boende (78% and 74%, respectively) coincided with other estimates reported in the literature (74–81% [*Muyembe and Kipasa, 1995*; *Khan et al., 1999*; *Ndambi et al., 1999*; *Sadek et al., 1999*; *Chowell et al., 2004*] and 74% [*Maganga et al., 2014*], respectively). The remaining outbreak estimates have not been studied by other authors and are reported here for the first time.

**Table 3**. Odds of dying from EVD

|  | OR | 2.50% | 97.50% |
| --- | --- | --- | --- |
| Outbreak Isiro 2012 | 0.89 | 0.31 | 2.59 |
| Outbreak Kikwit 1995 | 5.44 | 1.43 | 21.87 |
| Age [0, 5) | 1.12 | 0.23 | 6.63 |
| Age [5, 15) | 0.2 | 0.05 | 0.7 |
| Months since first case | 0.69 | 0.48 | 0.97 |
| Delay onset to hospitalisation | 1.11 | 1.02 | 1.21 |

EVD, Ebola virus disease.
Estimated through binomial regression with age group and year of outbreak as factorial covariates and the number of months since the start of the outbreak and the delay from symptom onset to hospitalisation as continuous covariates.

Overall, CFRs and delays between symptom onset and hospitalisation, symptom onset and death, and hospitalisation and death reported in our study do not differ substantially with those reported for the current outbreak (*WHO Ebola Response Team et al., 2015b*). The data presented were originally collected for the containment of the outbreaks rather than for providing the basis of an epidemiological study of the disease. As such, variables are not recorded consistently across all outbreaks and there are missing data. This dataset does not take into consideration undetected cases. A surveillance study carried out in northwestern DRC between 1981 and 1985, through clinical records and serological testing, detected 21 cases likely to be EVD, suggesting that sporadic cases do occur (*Jezek et al., 1999*). Another serosurvey carried out in Yambuku after the outbreak suggested that that 17% of the population in the village was infected asymptomatically (*Breman et al., 1978*). Under-reporting may differ between and during outbreaks and may impact the calculated estimates such as CFRs, which limits the validity of direct comparisons of values between outbreaks. Other limitations include the different case definitions employed in different outbreaks and that the method used to calculate the effective reproduction numbers is susceptible to changes in reporting during the outbreak (as most methods are). However, it is robust if the extent of underreporting remains constant during each outbreak. Moreover, it is robust to different reporting sensitivity between outbreaks.

The regular re-emergence of EVD in human hosts is likely to be connected to the presence of the virus in animal reservoirs, such as bats and monkeys (*Leroy et al., 2009*; *Muyembe-Tamfum et al., 2012*). The presence of vast tropical rainforests covering entire regions of the DRC and the strong link existing between local economies and the forest makes a re-emergence of the virus in the country in the near future very likely (*Pigott et al., 2014*). Although the Mweka 2007 outbreak has been linked to the consumption of fruit bats that migrate to the area (*Leroy et al., 2000*), the epidemiological link between index cases (when known) and animal reservoirs has not been found for any of these outbreaks.

All outbreaks except for the 2007 Mweka outbreak have involved hospital transmission during the early part of the outbreak, illustrating the amplifying effect that poor infection control can have on EVD epidemics. A study of the 1976 outbreak has highlighted the importance of community infection to transmission (*Camacho et al., 2014*). Traditional burials are an important mechanism of transmission of EVD. Funeral data can help inform mathematical models that explore the importance of this route of transmission and can help guide resource allocation. This will be explored in subsequent analysis.

Mweka 2008 was the shortest and smallest outbreak with the lowest CFR. This could be due to the short delay between the first notification and the opening of the isolation centre (10 days). The low CFR during Isiro could be due to infection by a less virulent type of virus (*B. ebolavirus*) and is in line with what has been reported for this virus in other outbreaks (*Van Kerkhove et al., 2015*).

In most outbreaks, major interventions arrived when the reproduction number, R, was less than

**Table 4**. Odds of dying from EVD

| Covariates | OR | 2.50% | 97.50% |
| --- | --- | --- | --- |
| Outbreak Isiro 2012 | 0.67 | 0.29 | 1.55 |
| Outbreak Kikwit 1995 | 4.63 | 1.96 | 10.94 |
| Outbreak Mweka 2007 | 3.83 | 1.61 | 9.18 |
| Outbreak Mweka 2008 | 0.25 | 0.1 | 0.63 |
| Outbreak Yambuku 1976 | 7.11 | 3.13 | 16.75 |
| Age [0, 5) | 2.49 | 1.12 | 6.34 |
| Age [5, 15) | 0.36 | 0.21 | 0.63 |
| Months since first case | 0.65 | 0.54 | 0.77 |

EVD, Ebola virus disease.
Estimated through binomial regression with age group and year of outbreak as factorial covariates and the number of months since the start of the outbreak as continuous covariate.

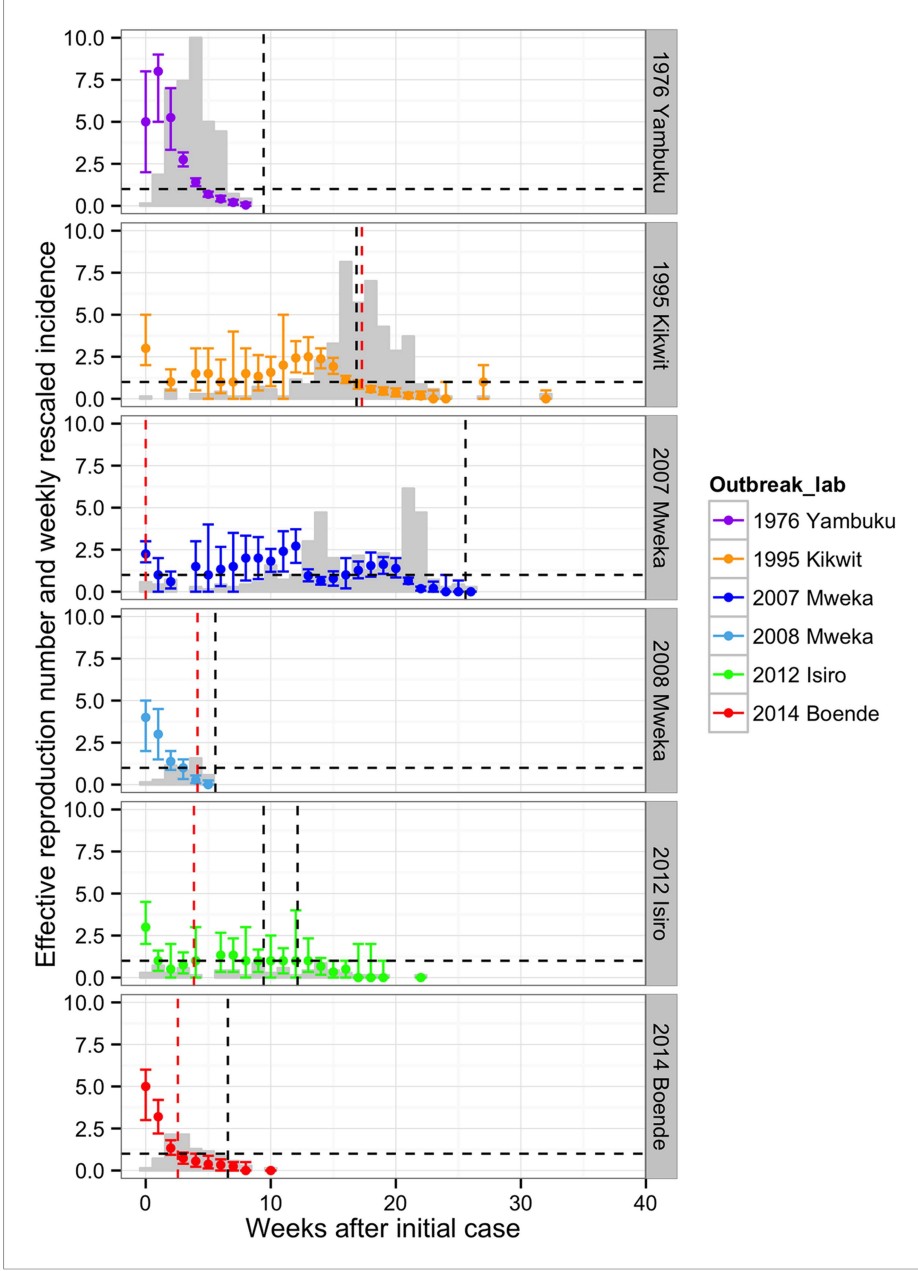

**Figure 6**. Evolving effective reproduction numbers with time after the start of the outbreak and adjusted weekly incidence. Weekly point estimates of the effective reproduction numbers are presented with 95% confidence intervals. The dashed horizontal line indicates the threshold R = 1. The vertical dashed lines represent important events that occurred during the outbreaks (in red, the first notifications, and in black, important interventions carried out). For Yambuku, this was the closure of Yambuku Mission Hospital; for Kikwit, the closure of all hospitals, health centres, and laboratories; for Mweka 2007, the opening of two mobile laboratories; for Mweka 2008, the opening of the first isolation centre; for Isiro, first the opening of the isolation centre and later the opening of the laboratory; and for Boende, the opening of the first isolation centre. The light grey bars represent the weekly incidence of Ebola virus disease (EVD) (omitting suspected cases) rescaled by dividing by seven.

one and the epidemic was already under control. This suggests an important role of other factors, such as changes in contact behaviour, in shaping the changes of R. For example, there is evidence that an increase in the proportion of patients admitted to hospital was associated with a reduction in the size of EVD transmission chains in Guinea in 2014 (*Faye et al., 2015*) and the community acceptance

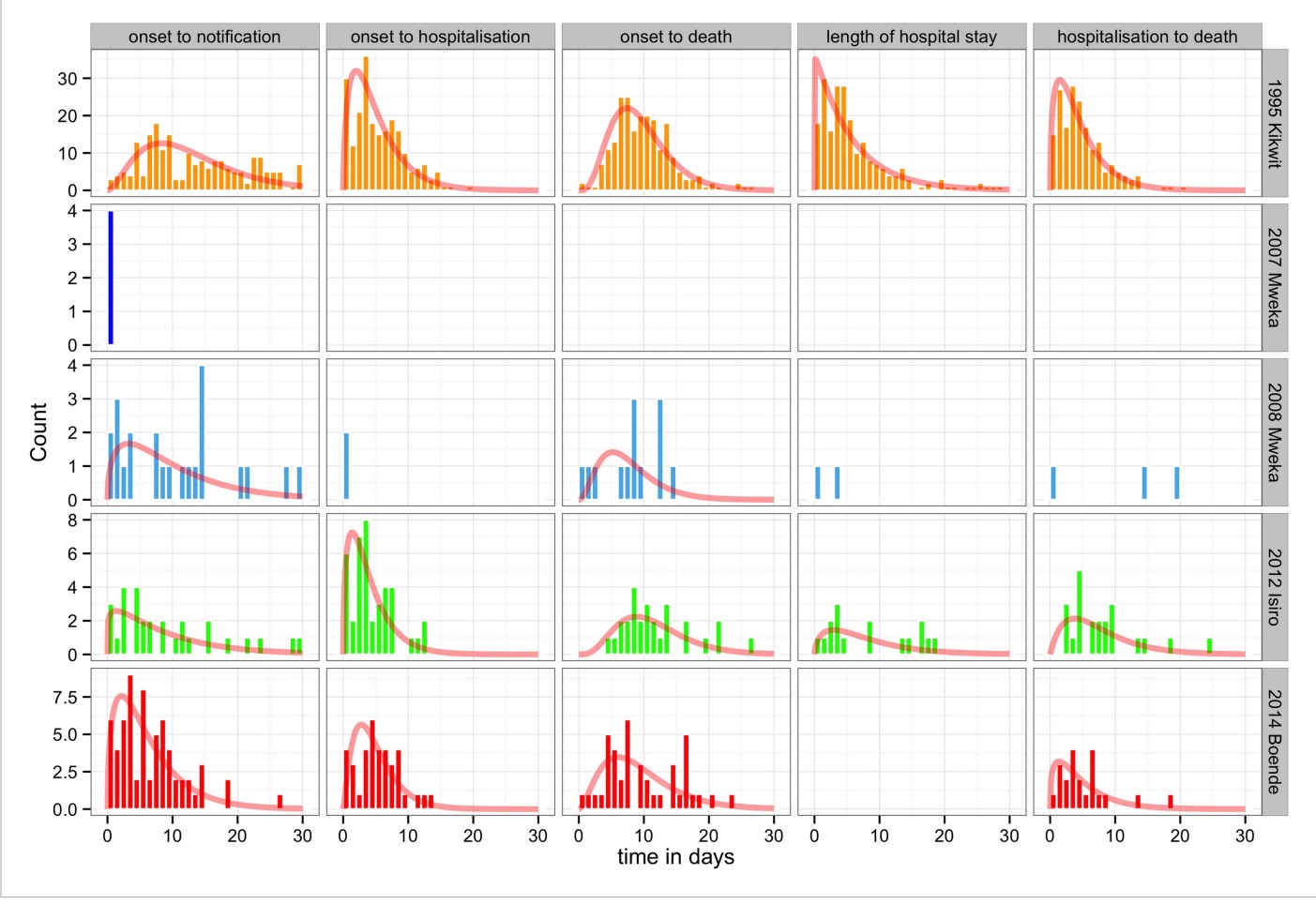

**Figure 7**. Delay distributions for the EVD outbreaks in the DRC. The bars represent the observed frequency distributions of the delay from onset of symptoms to notification, onset of symptoms to hospitalisation, onset of symptoms to death, length of hospitalisation, and date of hospitalisation to death. Delays were censored at 30 days. The red line represents the respective fit of a gamma distribution.

of EVD control measures in West Africa improved dramatically over the course of the epidemic, which led to better infection control (*Dhillon and Kelly, 2015*).

The Boende outbreak began whilst the West African outbreak was gaining international importance. This much smaller outbreak, with an initial R of five, which consisted of 68 cases, lasted only 10 weeks. The more remote setting, a background antibody presence in the area and a greater preparedness to EVD (that led to its notification 3 weeks after the first case and the opening of the first isolation centre a month later) could have contributed to the avoidance of a larger outbreak (*Heymann et al., 1980*; *Busico et al., 1999*; *Maganga et al., 2014*).

The high number of EVD cases between 25 and 64 years of age compared to the background demographics, the high CFR in children under five, the decrease in CFRs in those 5 to 15, and the subsequent increase in CFR during adulthood are phenomena that warrant further investigation. The variation in symptoms reported during different outbreaks is also a matter for further research.

## Materials and methods

### Data

Line list data and reports for each outbreak were retrieved from the Direction de Lutte contre la Maladie (DLM) (*Ministère de la Santé Publique (Direction de la Lutte contre la Maladie), 2007*, *2009*, *2012*; *Ministère de la Santé Publique (Comité National de Coordination), 2014*). The DLM is

**Table 5.** Mean values and standard deviations corresponding to the delay distributions

| Outbreak | Delay | Mean | SD |
|---|---|---|---|
| 1995 Kikwit | Onset to notification | 12.91 | 7.72 |
| 1995 Kikwit | Onset to hospitalisation | 5.02 | 3.91 |
| 1995 Kikwit | Onset to death | 9.47 | 4.44 |
| 1995 Kikwit | Length of hospital stay | 5.72 | 5.67 |
| 1995 Kikwit | Hospitalisation to death | 4.5 | 3.69 |
| 2007 Mweka | Onset to notification | 0 | 0 |
| 2008 Mweka | Onset to notification | 10.04 | 8.47 |
| 2008 Mweka | Onset to hospitalisation | 0 | 0 |
| 2008 Mweka | Onset to death | 7.62 | 4.44 |
| 2008 Mweka | Length of hospital stay | 1.5 | 2.12 |
| 2008 Mweka | Hospitalisation to death | 11 | 9.85 |
| 2012 Isiro | Onset to notification | 8.83 | 8.29 |
| 2012 Isiro | Onset to hospitalisation | 4 | 3.27 |
| 2012 Isiro | Onset to death | 11.37 | 5.41 |
| 2012 Isiro | Length of hospital stay | 8 | 6.56 |
| 2012 Isiro | Hospitalisation to death | 7.59 | 5.52 |
| 2014 Boende | Onset to notification | 6.23 | 5.08 |
| 2014 Boende | Onset to hospitalisation | 4.95 | 3.32 |
| 2014 Boende | Onset to death | 9.39 | 5.67 |
| 2014 Boende | Hospitalisation to death | 4.86 | 4.23 |

Delays distributions (delay from onset of symptoms to notification, onset of symptoms to hospitalisation, onset of symptoms to death, length of hospitalisation and date of hospitalisation to death).

the public body in charge of containing EVD outbreaks in the DRC. These data were designed for outbreak containment rather than for epidemiological analysis; therefore, appropriate cleaning was undertaken. The fields selected were age, sex, date of symptom onset, date of hospitalisation, date of hospital discharge, outcome, case definition, date of notification (when the case was first reported to the DLM), date of death, occupation, fever, diarrhoea, abdominal pain, headache, vomiting, hiccups, and hemorrhagic symptoms. Where this information was not available, it was left blank. A unique ID was assigned to each patient in the dataset. The Tandala outbreak (1977) included only one reported case; therefore, only the context and history of this outbreak was analysed. We included 262 of the 318 cases reported in Yambuku (those for which these data were available). The aggregated line lists can be found in *Supplementary file 1*.

## Case definitions

According to the WHO EVD case definitions for outbreak settings; suspected cases are all individuals (alive or dead) who had a fever and had contact with a suspected, probable, or confirmed EVD case or a sick or dead animal; any individual with a fever and more than three additional EVD symptoms; or any person with unexplained bleeding or whose death is unexplained (*World Health Organization, 2014*). Probable cases are suspected cases that have a clear epidemiological link with a confirmed case. Confirmed cases are individuals who were tested positive via PCR. In the DRC setting, the case definitions employed varied somewhat between outbreaks (Appendix 1, Section A). Unless stated otherwise, where the case definitions distinguished susceptible cases from probable and confirmed cases, all estimates presented (CFRs, symptom delays, and reproduction numbers) were computed omitting suspected cases.

## Patient demographics, epidemic curves, and symptoms

DRC national demographics between 1975 and 2010 were used as reported by the UN Department of Economic and Social Affairs (*United Nations (Department of Economic and Social Affairs), 2013*). For temporal comparison of patient reports, we used the date of infection. When available, we used the date of symptom onset. When these were unavailable, hospitalisation dates were used instead. If these were also absent, the notification dates were used as proxy.

When calculating the proportion of confirmed and probable cases that presented with EVD symptoms, we assumed that patients for whom the presence or absence of at least one symptom was reported did not display any additional symptoms unless those were also reported.

## CFRs, reproduction numbers, and delay distributions

The odds of dying from EVD were estimated through binomial regression with age group and year of outbreak as factorial covariates and the number of months since the start of the outbreak and the delay from symptom onset to hospitalisation as continuous covariates. The age groups used were 0–5 years, 5–15 years, and >15 years. The delay from symptom onset to hospitalisation was present for

63% of probable or confirmed cases. These dates were not recorded for Yambuku and Mweka 2007 and only for four cases for Mweka 2008. For this reason, these outbreaks were excluded from this first analysis. A second regression model was conducted that excluded the delays from symptom onset to hospitalisation as an explanatory variable, enabling the use of data from all major outbreaks and increasing statistical power. The start of an outbreak was defined by the earliest onset of symptoms of any detected case. The CIs were calculated using profiled log-likelihood.

We calculated the weekly effective R, the average number of individuals that were infected by a typical EVD case during the period of infectiousness, by reconstructing the transmission tree of each outbreak on the basis of date of infection for each case (*Wallinga and Teunis, 2004*). To link a case to its most likely source, we assumed a serial interval of 15.3 days with a standard deviation of 9.3 days as reported during the current outbreak in West Africa (*Maganga et al., 2014*). Delays in care were only calculated for those outbreaks for which the necessary dates were recorded.

## Software

R-3.1.2 was used for the cleaning, analysis, and plotting of figures (*R Development Core Team, 2011*).

## Ethical approval

This study was approved by the LSHTM Research Ethics Committee (approval number PR/1541/1541).

## Role of the funding source

The funders had no role in the design, collection, analysis, and interpretation of data, or in the writing of the manuscript. The corresponding authors had full access to all the data and were responsible for the final decision to submit for publication.

## Acknowledgements

This study was funded by the Fischer Family Trust, the National Institute for Health Research Health Protection Research Unit in Immunisation at the London School of Hygiene and Tropical Medicine in partnership with Public Health England, the Research for Health in Humanitarian Crises Programme (managed by Research for Humanitarian Assistance, Grant 13165). The views expressed are those of the authors and not necessarily those of the funders. We'd like to thank all the community nurses and volunteers on the ground that collaborated with the DLM to collect this data amid challenging circumstances.

## Additional information

### Funding

| Funder | Grant reference | Author |
|---|---|---|
| Fischer Family Trust | | Alicia Rosello, Mathias Mossoko, Placide Mbala, Marc Baguelin, Jean-Jacques Muyembe Tamfum |
| National Institute for Health Research Health Protection Research Unit in Immunisation at the London School of Hygiene and Tropical Medicine in partnership with Public Health England | EPIDZD03 | Albert Jan Van Hoek, Marc Baguelin |
| Research for Health in Humanitarian Crises | 13165 | Stefan Flasche, Anton Camacho, Sebastian Funk, Adam Kucharski |

The funders had no role in study design, data collection and interpretation, or the decision to submit the work for publication.

## Author contributions

AR, MB, Conception and design, Acquisition of data, Analysis and interpretation of data, Drafting or revising the article; MM, PM, BKI, Acquisition of data, Drafting or revising the article; SF, AJVH, AC, SF, AK, Analysis and interpretation of data, Drafting or revising the article; WJE, PP, J-JMT, Conception and design, Analysis and interpretation of data, Drafting or revising the article

## Ethics

Human subjects: This study was approved by the LSHTM Research Ethics Committee (approval number PR/1541/1541). Informed consent and consent to publish were not required as the data had no personally identifiable information.

# Additional files

## Supplementary files

• Supplementary file 1.  Aggregated line list for all outbreaks in the Democratic Republic of the Congo.

• Supplementary file 2.  Repository of the interventions carried out during all outbreaks in the Democratic Republic of the Congo.

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

## Appendix 1

# Section A. Case definitions for the different outbreaks

### Yambuku 1976 case definitions

## Suspected
All those presenting with headache and/or fever for at least 24 hr that had contact with a probable or confirmed case within the previous 21 days.

## Probable
Person living in the epidemic area who died after one or more days with two or more of the symptoms (headache, fever, abdominal pain, nausea and/or vomiting, and bleeding). The patient must have within the prior 21 days received an injection or had contact with a probable or confirmed case and the illness must not have any other obvious cause.

## Confirmed
Ebola virus isolated, demonstrated by electron microscopy or IFA titre of at least 1:64 within 21 days after symptom onset.

### Kikwit 1995 case definitions

## Probable
• Fever+ contact with Ebola virus disease (EVD) case.

OR

• Fever AND at least 3 symptoms (headache, vomiting/nausea, anorexia, diarrhoea, fatigue, abdominal pain, myalgia, generalized generalised joint pains, dysphagia, hiccups).

OR

• Any unexplained death.

## Clinical cases
All those presenting hemorrhagic symptoms.

### Mweka 2007, Mweka 2008/2009, and Isiro 2012 case definitions

## Suspected
• All those (dead or alive) presenting with fever that had contact with a patient suffering from a hemorrhagic fever or any sickly live or dead animal.

OR

• All those (dead or alive) presenting with fever and any three of the following symptoms:

| | |
|---|---|
| Headache | Vomiting |
| Anorexia/loss of appetite | Diarrhoea |
| Intense fatigue | Abdominal pain |
| Muscular or joint pain | Difficulty swallowing |
| Difficulty breathing | Hiccups |

OR

• Any unexplained bleeding.

OR

• Any unexplained death.

### Probable

- Suspected case evaluated by a clinician.

OR

- Deceased case with an epidemiological link to a confirmed case.

OR

- Suspected case that is known to have travelled in an area affected by the outbreak.

### Confirmed

Suspected or probable case that has been tested positive in the laboratory.

## Boende 2014 case definitions

### Suspected

- All those (dead or alive) presenting or having presented with a sudden-onset high fever that had contact with a suspected, probable or confirmed EVD case or any sickly live or dead animal.

OR

- All those (dead or alive) presenting or having presented with a sudden-onset high fever of sudden onset and any three of the following symptoms:

| Headache | Vomiting |
|---|---|
| Anorexia/loss of appetite | Diarrhoea |
| Intense fatigue | Abdominal pain |
| Muscular or joint pain | Difficulty swallowing |
| Difficulty breathing | Hiccups |

OR

- Any unexplained bleeding.

OR

- Any unexplained sudden death.

### Probable

All suspected cases that cannot be confirmed biologically but have an epidemiological link with a confirmed case (as determined by the surveillance committee).

### Confirmed

Suspected or probable cases with a positive result in the laboratory (presence of Ebola antigen, presence of viral RNA detected by PCR or presence of anti-Ebola IgM).

### Non-case

All suspected cases with a negative laboratory result (no Ebola antigen, no RNA by PCR, and no specific anti-Ebola IgM detected).

## Section B. Description of the outbreaks

### Yambuku, 1976

The first known case was a 44-year-old male teacher who fell ill after a trip near Gbadolite and was admitted at the Yambuku Catholic Mission Hospital on the 26th of August 1976 with a febrile illness thought to be malaria/typhoid fever. He later developed gastrointestinal bleeding and died on the 8th of September. The hospital held 120 beds and was run by 17 medical staff including three Belgian nuns. Five syringes and needles were provided to the hospital outpatient workers every morning and these were re-used on different patients without appropriate sterilisation. Parenteral injection was the main delivery method for the majority of

the medication administered to patients in the hospital, including vitamins given to pregnant women, and greatly contributed to the spread of disease through this population. On the 30th of September, the Yambuku Mission Hospital was closed and the Ebola virus (Z. *ebolavirus*) was isolated on the 13th of October. Between the 1st of September and the 24th of October, 318 cases and 280 deaths were reported (*World Health Organization, 1978*).

## Tandala, 1977

The following year, in June 1977, a 9-year-old girl presented to the Tandala Mission Hospital also in the Equateur province, approximately 300 km west from Yambuku. She died of an acute hemorrhagic fever, later recognised as EVD (*Heymann et al., 1980*).

## Kikwit, 1995

In 1995, cases of EVD were reported in the city of Kikwit, the most densely populated area affected by EVD in the Democratic Republic of the Congo (DRC) to date, and home at the time to 200,000 residents (*Khan et al., 1999*; *Muyembe-Tamfum et al., 1999*). The index case was thought to be a charcoal maker who worked in the forest close to Kikwit. The first cases were observed in early April at the Kikwit II Maternity Hospital. After a laboratory technician from this facility was transferred to Kikwit General Hospital and underwent two laparotomies for a suspected perforated bowel, two nurses assisting the procedure and several providing post-operative care became ill. At the end of April, an outbreak of bloody diarrhoea was reported in the Kikwit General Hospital. Initially, shigellosis was suspected, but upon receipt of laboratory supplies sent from Kinshasa, patient samples were tested and this hypothesis was discarded. Blood from 14 acutely ill and convalescent patients was collected on the 4th of May and sent for testing to the CDC, Atlanta. All hospitals, health centres, and laboratories in the Kikwit area were closed. Patients were quarantined at the Kikwit General Hospital with no running water, electricity, nor latrines. One physician and three nurses volunteered to stay with the patients. On the 10th of May, the CDC confirmed the EVD diagnosis. The quarantine of patients in the Kikwit General Hospital had limited success because food had to be provided by patients' relatives. In addition, due to the poor living conditions, many patients fled the hospital (*Muyembe-Tamfum et al., 1999*). Between the 13th of January and the end of June, 317 cases and 248 deaths were reported.

## Mweka, 2007

The first cases of EVD during the Mweka outbreak in 2007 were detected in the Mweka, Bulape, and Luebo health zones of the Kasai Occidental province. The Mweka health zone alone spans over 20,000 km and comprises over 170,000 inhabitants. These cases were initially attributed to a novel disease originally named the Kampungu syndrome, a hybrid between typhoid fever, shigellosis, and Ebola. However, it was later recognised that three separate outbreaks of these diseases were co-existing in the same location. The retrospective nature of the diagnosis hindered the epidemiological investigation. From April to October, 264 cases and 187 deaths were reported. However, only 24 cases were confirmed. Two international mobile laboratories were opened on the 27th of September, one in Mweka (Public Health Agency of Canada, PHAC) and one in Luebo (CDC) (*Ministère de la Santé Publique (Direction de la Lutte contre la Maladie), 2007*). This outbreak was later associated with the consumption of fruit bats that annually migrate in masses to the area (*Leroy et al., 2009*).

## Mweka, 2008/2009

One year later, a new outbreak of EVD was detected in the same area. The epicentre of the outbreak was Kaluamba (Mweka) but the outbreak spread into the neighbouring district of Luebo. The first reported case was an 18-year-old mother who had been suffering from fever since the 15th of November and was hospitalized at the Kaluamba health centre on the 27th. She died from a post-partum haemorrhage after giving birth to a premature 6-month-old baby who died the same day. A traditional burial was then conducted. Several of her contacts including the nurse that saw her at the health centre developed EVD. The epidemiologists

conducting the investigation postulated that she could have been infected though the injections she had received from the 4th to the 11th of November by a clandestine nurse from Kaluamba in an attempt to decrease her fever. This clandestine nurse had previously treated another patient who had died 2 weeks before. Between the 18th and the 25th of December, national and international support arrived on site and MSF Belgium volunteers opened the first isolation centre in Kaluamba on the 27th of December. On the 2nd of January, the isolation centre was transferred from Kaluamba to Kampungu, where new cases were being reported and MSF had built an isolation centre during the preceding outbreak with greater capacity (25 patients). Three mobile laboratories were opened on the 15th of January (*Ministère de la Santé Publique (Direction de la Lutte contre la Maladie), 2009*). In total, from November to January, 32 cases and 14 deaths were reported.

## Isiro, 2012

On the 12th of July 2012, an EVD outbreak was declared in the Kibaale region of Uganda, east of the Albert Lake. On the other side of this lake lies the Orientale province of the DRC. Due to the proximity to the outbreak, the DRC reinforced EVD surveillance in this region. On the 2nd of August, the first suspected cases were reported in the Isiro and Dungu health zones in the Haut-Uélé (a population estimated at 700,000), making this the first known cross-border epidemic. Already in June, seven HCWs had died at the Chemin de Fer des Uélé clinic (Isiro) from a gastrointestinal febrile illness. When, in July, a daughter and her mother (who was caring for her) at the Isiro General Reference Hospital presented with a similar illness with additional hemorrhagic symptoms, samples were sent to the CDC's Uganda Virus Research Institute laboratory in Entebbe. The diagnosis of EVD was confirmed on the 16th of July. This time, the infecting pathogen was the *B. ebolavirus*, different from the *Z. ebolavirus*, encountered in all other outbreaks in the DRC. On the 6th of August, a treatment centre was established at the Isiro General Reference Hospital that was re-organised by MSF Belgium and Spain who arrived on site on the 10th of August. On the 25th of August, the first laboratory was installed. The last case was confirmed on the 11th of October and the outbreak was declared over on the 24th of November. In total, 52 cases and 28 deaths were reported (*Ministère de la Santé Publique (Direction de la Lutte contre la Maladie), 2012*).

## Boende, 2014

The last outbreak to date took place 2 years later in the Boende health zone in the Djera sector (Equateur province), which comprises approximately 250,000 inhabitants. In mid-August 2014, a set of suspiciously grouped cases and deaths were reported by the military personnel of the Watsi-Kengo camp to the central government. A 34-year-old pregnant woman from the Ikanamongo village who was receiving antenatal care in her village started to develop a fever in the first week of August. She was transferred to the Miracle centre in Isaka but died a few hours after her arrival. Because it is traditional not to burry a heavily pregnant mother with her child still inside her, a post-mortem cesarean section was carried out. All the HCWs participating in the procedure later died of EVD. Days before these events, the nurse designated to the Lokolia health centre had reported cases of a disease resembling Ebola to the Central Bureau but was told to keep quiet about his suspicions. On the 18th of August, the health area supervisor reported similar findings to that of the military to the Direction de Lutte contre la Maladie (DLM). Eight blood samples were collected from patients hospitalised in Lokolia and Watsi-Kengo and sent to the INRB, Kinshasa. Six were found positive to *Z. ebolavirus*. The outbreak was officially declared over on the 21st of (*Ministère de la Santé Publique (Comité National de Coordination), 2014*). 68 cases and 49 deaths were reported.

