## [Decision Letter]

Thank you for submitting your work entitled “Ebola Virus Disease in the Democratic Republic of the Congo: 1976-2014” for peer review at *eLife*. Your submission has been favorably evaluated by Prabhat Jha (Senior Editor), a Reviewing Editor, and two reviewers, one of whom, Simon Hay, has agreed to reveal his identity.

The reviewers have discussed the reviews with one another and the Reviewing editor has drafted this decision to help you prepare a revised submission.

Democratic Republic of the Congo (DRC) is the country that experienced the largest number of Ebola outbreaks since Ebola was discovered in 1976. The authors present a near complete line list for all reported outbreaks of Ebola virus disease (EVD) in the DRC and discuss some of the key epidemiological parameters calculated from this data, investigating how some of these have changed over the decades. The data linked with this paper is a fantastic resource and unique in terms of comprehensiveness and temporal spread – it is the only open-access Ebola resource of this type and is likely the only one that comes close to the datasets that are currently being used for the West African outbreak. This paper is an important contribution to the literature.

Essential revisions:

It is slightly disappointing that although the authors promise to “provide improved insights into EVD” (Introduction), the subsequent analyses performed on the data are relatively limited in scope. The authors need to consider the following in their revisions:

1) As a result of the West African outbreak, several studies exist that review related metrics most notably those published in Scientific Data from authors at Imperial (“A review of epidemiological parameters from Ebola outbreaks to inform early public health decision-making”). Whilst this study is sufficiently different to stand apart from this, it might be worth the authors considering drawing in information from other sources as a point of comparison/contrast.

2) The limitations of the approach used must be included in the Discussion. Whilst some caveats are provided (second paragraph), more details and consideration on the impact this may have should be considered, particularly when these may have inconsistent effects over the different outbreaks and time periods. For instance, under-reporting of cases will have a considerable impact on the case-fatality rates. The trends the authors report could therefore be just as much to do with poor healthcare access in more remote locations, improving diagnostic capacity over time in different areas and increasing recognition that Ebola virus disease can present in more milder forms (hence the name change from Ebola hemorrhagic fever). This appreciation for milder symptoms/possible asymptomatics might mean that some of the older outbreaks had significant underreporting. A formal quantitative analysis (similar to some of the work done for the current outbreak by the WHO working group) is likely not feasible on this dataset, however a more detailed consideration on why direct comparisons of values from each outbreak may be misleading is warranted. This will likely impact all the metrics that the paper discusses and will likely limit the inferences that can be made from this data.

3) There has been a lot of work on epidemiological parameters of EVD outbreaks and this work needs to be referenced more fully. The SciData review of epidemiological parameters provides a variety of different measures also calculated in this article – it would be interesting to see how this work differs/reinforces previous estimates from different outbreaks. For instance, do the assertions made in the fourth paragraph of the Discussion on CFR linked to species and notification date seem present in other outbreaks? The age trend in CFR is interesting to note. The authors suggest comparison with the West African outbreak (fifth paragraph). There is an interesting comparison that can be made between this data and the West African dataset (see the NEJM paper “Ebola virus disease among Children in West Africa”), which shows similar trends occurring between these two different areas. Similarly, tying the study to synoptic assessments of West Africa in general (e.g. “West African Ebola epidemic after one Year – slowing but not yet under control”) would be good to see.

4) Overall the dataset provided and the discussion of some of the trends between these different outbreaks is an important contribution to the field. The manuscript can be enhanced taking into account the points above.

---

## [Author Response]

Essential revisions:

It is slightly disappointing that although the authors promise to “provide improved insights into EVD” (Introduction), the subsequent analyses performed on the data are relatively limited in scope. The authors need to consider the following in their revisions:

1) As a result of the West African outbreak, several studies exist that review related metrics most notably those published in Scientific Data from authors at Imperial (“A review of epidemiological parameters from Ebola outbreaks to inform early public health decision-making”). Whilst this study is sufficiently different to stand apart from this, it might be worth the authors considering drawing in information from other sources as a point of comparison/contrast.

This is a pertinent remark, which we have addressed in the Discussion (third paragraph). Most of the estimates presented in this article are for outbreaks that have not been studied before. Where this is not the case, our estimates by and large coincide with those reported in the literature.

2) The limitations of the approach used must be included in the Discussion. Whilst some caveats are provided (second paragraph), more details and consideration on the impact this may have should be considered, particularly when these may have inconsistent effects over the different outbreaks and time periods. For instance, under-reporting of cases will have a considerable impact on the case-fatality rates. The trends the authors report could therefore be just as much to do with poor healthcare access in more remote locations, improving diagnostic capacity over time in different areas and increasing recognition that Ebola virus disease can present in more milder forms (hence the name change from Ebola hemorrhagic fever). This appreciation for milder symptoms/possible asymptomatics might mean that some of the older outbreaks had significant underreporting. A formal quantitative analysis (similar to some of the work done for the current outbreak by the WHO working group) is likely not feasible on this dataset, however a more detailed consideration on why direct comparisons of values from each outbreak may be misleading is warranted. This will likely impact all the metrics that the paper discusses and will likely limit the inferences that can be made from this data.

This is a valid limitation that we have incorporated into the Discussion (fourth paragraph).

3) There has been a lot of work on epidemiological parameters of EVD outbreaks and this work needs to be referenced more fully. The SciData review of epidemiological parameters provides a variety of different measures also calculated in this article – it would be interesting to see how this work differs/reinforces previous estimates from different outbreaks. For instance, do the assertions made in the fourth paragraph of the Discussion on CFR linked to species and notification date seem present in other outbreaks?

Unfortunately there is no information on the delay between first notification and the opening of the isolation centre or other similar measures taken in other *Bundibugyo ebolavirus* outbreaks but we have referred to the SciData review for comparison in the Discussion of CFRs during *Bundibugyo ebolavirus* outbreaks (seventh paragraph).

The age trend in CFR is interesting to note. The authors suggest comparison with the West African outbreak (fifth paragraph). There is an interesting comparison that can be made between this data and the West African dataset (see the NEJM paper “Ebola virus disease among Children in West Africa”), which shows similar trends occurring between these two different areas. Similarly, tying the study to synoptic assessments of West Africa in general (e.g. “West African Ebola epidemic after one Year – slowing but not yet under control”) would be good to see.

We have added comparisons to the “Ebola virus disease among Children in West Africa” paper in the Discussion (first paragraph) and also cited the “West African Ebola epidemic after one Year – slowing but not yet under control” paper you mention (fourth paragraph).